# ProtoGNN: Prototype-Conditioned Graph Refinement for Meaningful RNA–Protein Interaction Representations

## Abstract

We study RNA–protein interaction (RPI) prediction in a setting where each RNA and protein is represented only by fixed pretrained sequence embeddings. We propose **ProtoGNN**, a bipartite graph refinement model that augments standard edge scoring with *prototype-conditioned, contrastive-inspired* logit regularization. ProtoGNN refines node states via streamlined bipartite propagation with adaptive raw→graph fusion, constructs type-specific prototypes, and uses them to provide pair-specific global context during scoring. We also introduce a strong embedding-only baseline, **PairMLP**, to quantify how much signal is present in the pretrained representations alone. Across NPInter2 and RPI7317 under 5-fold edge-level cross-validation, ProtoGNN consistently improves over PairMLP and matches or modestly improves upon previously reported baselines (including ZH-MolGraph) under the same benchmark setting, achieving MCC 0.9191 on NPInter2 and 0.8387 on RPI7317. To assess whether the learned representations are useful beyond binary decisions, we evaluate RNA-to-protein retrieval (RBP identification): given an RNA query, the model ranks all proteins by predicted interaction probability. On NPInter2, ProtoGNN achieves MRR $\approx 0.30$ with Recall@10 $\approx 0.26$, Recall@20 $\approx 0.51$, and Recall@50 $\approx 0.73$, indicating that refined fixed embeddings can support shortlist-style candidate prioritization.

## 1 Introduction

Ribonucleic acids (RNAs) play diverse and critical roles in cellular biology by interacting with proteins, other RNAs, and small molecules (Chen, 2020; Mou et al., 2022). Understanding RNA–protein interactions (RPIs) is essential for clarifying post-transcriptional gene regulation and disease mechanisms (Statello et al., 2021; Cramer, 2019; Weidmann et al., 2021). For example, disruptions in RNA–protein binding can contribute to dysregulation in cancer and other diseases, and many viral lifecycles depend on specific RNA–protein contacts. Experimental identification of RPI pairs, however, is laborious and time-consuming, involving techniques such as CLIP-seq or structural assays (Danan et al., 2016; Darnell, 2010). This motivates reliable computational RPI prediction to guide biological discovery.

Early computational methods for RPI relied on hand-crafted sequence features and conventional classifiers. *RPISeq* pioneered this approach by encoding RNAs and proteins as k-mer frequency vectors and applying Random Forest or SVM to classify interactions (Muppirala et al., 2011). Subsequent methods such as *IPMiner* used deep neural networks (stacked autoencoders) to learn latent features from k-mer profiles (Pan et al., 2016). Graph-based approaches also emerged: *NPI-GNN* formulates RPI prediction as link prediction in a bipartite interaction graph and applies a GNN with subgraph pooling (SEAL) to capture network topology (Shen et al., 2021). Another deep learning method, *RPITER*, introduced a hierarchical architecture that integrates multiple feature streams for ncRNA–protein interaction prediction (Peng et al., 2019). While these methods made progress, many depended on engineered features or interaction-network heuristics, and robustness on harder settings remains an open issue.

Recently, biological language models have enabled a new paradigm for RPI prediction. Large-scale pretrained models such as *RNA-FM* for RNAs and *ProtT5* for proteins learn contextual representa-

tions from unlabeled sequence corpora (Elnaggar et al., 2021; Chen et al., 2022). Incorporating such embeddings can capture useful biochemical and structural cues without explicit feature design. *ZH-MolGraph* combined pretrained RNA/protein embeddings with graph-based refinement over known RPIs, reporting strong performance on standard benchmarks (Liu et al., 2025). In parallel, *BioLLM-Net* explored sequence-only interaction modeling with gating-based fusion across interaction types (Abir et al., 2025). Together, these works highlight that pretrained sequence embeddings provide a strong foundation for RPI prediction; however, there is still room to study *how* to best exploit these embeddings with simple, reproducible architectures and clear ablations.

Importantly, strong classification metrics alone do not fully capture how embedding-based models behave in a practical discovery workflow. In many biological settings, one queries an RNA and seeks to prioritize a short list of likely RNA-binding proteins (RBPs) for follow-up validation. This motivates evaluating *RNA-to-protein retrieval*: ranking all proteins for a given RNA by predicted interaction score, which directly tests whether the learned representations provide *useful ordering* of candidates rather than only thresholded decisions.

In this work, we propose **ProtoGNN**, a prototype-conditioned graph refinement model for RPI prediction built on fixed RNA/protein embeddings. ProtoGNN uses a bipartite message-passing backbone to refine node representations from training positives, and introduces (i) degree-aware fusion to control how much each node moves from its raw embedding to its graph-refined embedding, and (ii) a prototype-conditioned scoring mechanism that uses learnable type-specific prototypes to supply global context during pair scoring. Importantly, our goal is not to change the embedding pipeline; rather, we ask whether a small set of architectural choices around fixed embeddings can yield consistent improvements over a strong embedding-only baseline under the same evaluation protocol. To quantify the contribution of graph refinement beyond embeddings, we also introduce **PairMLP**, a simple pairwise MLP baseline operating solely on the fixed RNA/protein embeddings. We further assess *RBP identification* via RNA-centric retrieval metrics (MRR, AP, Recall@K/Prec@K), providing a more usage-aligned evaluation of how refined representations support candidate prioritization.

OUR CONTRIBUTIONS

- **Prototype-conditioned graph refinement for RPI:** We propose ProtoGNN, combining bipartite message passing with degree-aware fusion and type-specific prototypes for pair scoring.

- **Strong embedding-only baseline:** We report PairMLP as a transparent baseline under the same fixed-embedding setting, showing that pretrained embeddings already provide substantial predictive signal.

- **Systematic evaluation:** We evaluate on NPInter2 and RPI7317 with 5-fold cross-validation and provide ablations to isolate which design choices contribute to performance.

- **Meaningful retrieval behavior:** We test whether the learned representations support RNA→protein candidate prioritization (RBP identification), reporting retrieval metrics (MRR/AP/Recall@K) and case studies.

The rest of the paper is organized as follows. Section 2 reviews background and related work on RPI prediction and embedding-based modeling. Section 3 presents our methods, including PairMLP and ProtoGNN. Section 4 reports results and ablations, while Section 5 discusses key findings, limitations, and future work.

## 2 RELATED WORK

**Traditional RPI prediction.** Early computational methods relied on manually designed sequence descriptors and classical classifiers. *RPISeq* (Muppirala et al., 2011) represents RNAs and proteins using k-mer composition vectors and trains Random Forest or SVM models. Such approaches are simple and efficient, but their representational capacity is limited by fixed, local feature templates. Other lines of work incorporated additional biological priors (e.g., domains or interaction networks), but performance is often sensitive to feature choice and data coverage.

**Deep learning and graph-based models.** Neural approaches improved feature learning from sequence-derived inputs. *IPMiner* (Pan et al., 2016) used stacked autoencoders to learn latent representations from k-mer profiles, improving over earlier baselines while still depending on hand-crafted sequence statistics. Graph-based methods such as *NPI-GNN* (Shen et al., 2021) cast RPI prediction as link prediction in an interaction graph, extracting local enclosing subgraphs (SEAL) and applying GNNs to capture topological motifs. *RPITER* (Peng et al., 2019) proposed a hierarchical architecture integrating multiple neural components for ncRNA–protein interaction prediction. These approaches highlight the value of learned representations and network structure, but graph reliance can be challenging when interaction evidence is sparse or unevenly distributed.

**Pretrained language model embeddings.** Large-scale pretrained models for biomolecules learn contextual sequence representations from massive unlabeled corpora. For proteins, ProtTrans models such as ProtT5 provide strong embeddings (Elnaggar et al., 2021), while RNA-FM provides analogous representations for RNAs (Chen et al., 2022). *ZHMolGraph* (Liu et al., 2025) combined such pretrained embeddings with graph-based refinement over an RPI network and reported strong benchmark results. These developments suggest that fixed embeddings can encode substantial biochemical signal, motivating careful study of architectures that operate directly on these representations.

**Sequence-only fusion.** In parallel, sequence-only interaction modeling explores how to combine pretrained embeddings without requiring explicit structural inputs. *BioLLMNet* (Abir et al., 2025) projects partner embeddings to a shared space and uses a gating mechanism to adaptively fuse modalities, demonstrating that effective fusion can be competitive in embedding-based interaction prediction settings.

**Positioning of our work.** ProtoGNN builds on the above trends by keeping the input pipeline fixed (pretrained RNA/protein embeddings) and focusing on architectural choices at the interaction-model level. Specifically, it couples simplified bipartite message passing with degree-aware fusion and a prototype-conditioned scoring mechanism, aiming to improve over a transparent embedding-only baseline under the same evaluation protocol.

## 3 METHODS

### 3.1 DATASET AND EMBEDDING PIPELINE

We evaluate on two benchmark RPI datasets: NPInter2 (10,412 interactions across 4,636 RNAs and 449 proteins) and RPI7317 (7,317 interactions compiled from multiple sources) (Liu et al., 2025). Since both datasets contain only verified positives, we use the released benchmark negatives constructed by random RNA–protein pairing to match the number of positives, consistent with prior protocols (Pan et al., 2016; Shen et al., 2021). All results use 5-fold stratified edge-level cross-validation. In addition to edge-level classification, we evaluate an *RNA-to-protein retrieval* setting that mirrors RBP identification: given a query RNA, the model ranks all candidate proteins by predicted interaction score.

We use the provided pretrained sequence embeddings distributed with the benchmark package. RNA sequences are embedded with RNA-FM into 640-dimensional vectors (Chen et al., 2022), and protein sequences are embedded with ProtTrans ProtT5-XL-UniRef50 into 1024-dimensional vectors (Elnaggar et al., 2021). Embeddings are fixed (no fine-tuning); all models project them into a shared latent space of dimension $d$.

### 3.2 PAIRMLP: A SIMPLE PAIRWISE MLP BASELINE

PairMLP is our simplest interaction model that uses *only* the fixed RNA/protein embeddings described in the previous subsection, without constructing or propagating over any interaction graph. Its purpose is to quantify how much predictive signal is already present in the pretrained sequence embeddings when combined with a standard pairwise classifier.

Let $\mathbf{x}_r \in \mathbb{R}^{d_r}$ and $\mathbf{x}_p \in \mathbb{R}^{d_p}$ denote the RNA and protein embeddings, respectively. We first project them to a common latent dimension $d$ using two independent encoders:

$$\mathbf{h}_r = f_r(\mathbf{x}_r), \qquad \mathbf{h}_p = f_p(\mathbf{x}_p), \tag{1}$$

where each encoder consists of a linear layer followed by ReLU activation and dropout, ensuring consistency with the overall architecture.

We then form a pair representation using four complementary interaction features:

$$\mathbf{z}_{rp} = [\mathbf{h}_r \,;\, \mathbf{h}_p \,;\, |\mathbf{h}_r - \mathbf{h}_p| \,;\, \mathbf{h}_r \odot \mathbf{h}_p] \in \mathbb{R}^{4d}, \tag{2}$$

where $|\cdot|$ is elementwise absolute difference and $\odot$ denotes the Hadamard product. Finally, an MLP head produces a scalar logit:

$$\ell_{rp} = g(\mathbf{z}_{rp}), \tag{3}$$

where $g(\cdot)$ is a 3-layer MLP with ReLU and dropout: $4d \rightarrow 512 \rightarrow 256 \rightarrow 1$.

### 3.3 PROTOGNN: PROTOTYPE-AUGMENTED CONTRASTIVE BIPARTITE GNN

ProtoGNN is designed for RNA–protein link prediction using only pretrained embeddings, while introducing three inductive biases: (i) simple bipartite propagation to refine node states, (ii) adaptive raw$\rightarrow$graph fusion via degree and pair-specific gates, and (iii) prototype-driven contrastive augmentation that improves separability.

Let $\mathcal{G} = (\mathcal{R} \cup \mathcal{P}, \mathcal{E})$ denote the bipartite training graph for a fold, with RNA nodes $\mathcal{R}$, protein nodes $\mathcal{P}$, and observed training interactions $\mathcal{E} \subseteq \mathcal{R} \times \mathcal{P}$. Let $\mathbf{A} \in \{0,1\}^{|\mathcal{R}| \times |\mathcal{P}|}$ be the (sparse) bipartite adjacency.

**(1) Modality-specific encoders.** We first project RNA and protein embeddings into a shared $d$-dimensional space using GELU and dropout:

$$\mathbf{h}_r^{(0)} = \mathrm{Drop}(\mathrm{GELU}(\mathbf{W}_r \mathbf{x}_r + \mathbf{b}_r)) \in \mathbb{R}^d, \tag{4}$$

$$\mathbf{h}_p^{(0)} = \mathrm{Drop}(\mathrm{GELU}(\mathbf{W}_p \mathbf{x}_p + \mathbf{b}_p)) \in \mathbb{R}^d. \tag{5}$$

**(2) Sparse edge dropout on the training graph.** During training, we apply sparse edge dropout on the bipartite adjacency:

$$\tilde{\mathbf{A}} = \mathrm{DropEdges}(\mathbf{A}, p), \tag{6}$$

where each nonzero edge is dropped with probability $p$ and remaining edge weights are re-scaled by $1/(1-p)$ to preserve expectation. This acts as graph-level regularization.

**(3) Bipartite propagation with residual mixers.** We perform $L$ layers of bipartite message passing using symmetric normalization:

$$\mathbf{m}_r^{(t)} = \mathbf{D}_r^{-1/2} \tilde{\mathbf{A}} \mathbf{D}_p^{-1/2} \mathbf{h}_p^{(t)}, \tag{7}$$

$$\mathbf{m}_p^{(t)} = \mathbf{D}_p^{-1/2} \tilde{\mathbf{A}}^\top \mathbf{D}_r^{-1/2} \mathbf{h}_r^{(t)}, \tag{8}$$

where $\mathbf{D}_r$ and $\mathbf{D}_p$ are degree matrices of the dropped adjacency $\tilde{\mathbf{A}}$. To stabilize training, we use a streamlined residual update with a per-layer mixer and LayerNorm:

$$\mathbf{h}_r^{(t+1)} = \mathrm{LN}\left(\mathbf{h}_r^{(t)} + \mathrm{Mix}_r^{(t)}(\mathbf{m}_r^{(t)})\right), \tag{9}$$

$$\mathbf{h}_p^{(t+1)} = \mathrm{LN}\left(\mathbf{h}_p^{(t)} + \mathrm{Mix}_p^{(t)}(\mathbf{m}_p^{(t)})\right), \tag{10}$$

where each $\mathrm{Mix}(\cdot)$ is a Linear$\rightarrow$GELU$\rightarrow$Drop block. In our experiments we use $L = 2$.

**(4) Degree-gated raw$\rightarrow$graph fusion.** To reduce over-smoothing and protect low-degree nodes, we fuse raw and propagated states using a learned degree gate computed from $\log(1 + \deg)$:

$$g_r = \sigma(\mathrm{MLP}_r(\log(1 + \deg_r))) \in (0, 1), \tag{11}$$

$$g_p = \sigma(\mathrm{MLP}_p(\log(1 + \deg_p))) \in (0, 1), \tag{12}$$

$$\mathbf{z}_r = \mathbf{h}_r^{(0)} + g_r \odot (\mathbf{h}_r^{(L)} - \mathbf{h}_r^{(0)}), \tag{13}$$

$$\mathbf{z}_p = \mathbf{h}_p^{(0)} + g_p \odot (\mathbf{h}_p^{(L)} - \mathbf{h}_p^{(0)}). \tag{14}$$

This lets the model interpolate between semantic (raw) and relational (graph-refined) representations.

**(5) Type-aware prototypes and edge-conditioned core.** ProtoGNN builds $K$ prototypes for RNAs and proteins from the degree-gated node embeddings $\{\mathbf{z}_r\}$ and $\{\mathbf{z}_p\}$ using learnable queries with soft assignment (softmax over nodes), yielding $\mathbf{P}_r \in \mathbb{R}^{K \times d}$ and $\mathbf{P}_p \in \mathbb{R}^{K \times d}$.

Given a candidate pair $(r, p)$, we compute an edge-conditioned "core" vector by attending over prototypes using the *pair-gated* representations (defined next):

$$\alpha_r = \text{softmax}\left(\frac{\mathbf{W}_q \bar{\mathbf{z}}_r \mathbf{P}_r^\top}{\sqrt{d}}\right) \in \mathbb{R}^K, \quad \alpha_p = \text{softmax}\left(\frac{\mathbf{W}_q \bar{\mathbf{z}}_p \mathbf{P}_p^\top}{\sqrt{d}}\right) \in \mathbb{R}^K, \tag{15}$$

$$\mathbf{c}_{rp} = \text{Fuse}([\alpha_r \mathbf{P}_r \; ; \; \alpha_p \mathbf{P}_p]) \in \mathbb{R}^d. \tag{16}$$

This yields a pair-specific summary derived from global prototype structure.

**(6) Pair-gated fusion for edge scoring.** Beyond degree gating, we use a *pair gate* that decides how much each specific edge should move from raw features to graph-refined features. For candidate edge $(r, p)$ we compute:

$$a_{rp} = \sigma\Big(\text{Gate}\left([\mathbf{h}_r^{(0)}; \mathbf{h}_p^{(0)}; |\Delta|; \odot]\right)\Big) \in (0, 1), \tag{17}$$

$$\bar{\mathbf{z}}_r = \mathbf{h}_r^{(0)} + a_{rp}(\mathbf{z}_r - \mathbf{h}_r^{(0)}), \quad \bar{\mathbf{z}}_p = \mathbf{h}_p^{(0)} + a_{rp}(\mathbf{z}_p - \mathbf{h}_p^{(0)}), \tag{18}$$

where $|\Delta| = |\mathbf{h}_r^{(0)} - \mathbf{h}_p^{(0)}|$ and $\odot = \mathbf{h}_r^{(0)} \odot \mathbf{h}_p^{(0)}$. This prevents over-reliance on propagation for pairs that are already separable from semantics alone.

**(7) Edge MLP + bilinear link term.** We score the pair using an MLP over interaction features plus a bilinear term:

$$\mathbf{f}_{rp} = [\bar{\mathbf{z}}_r; \bar{\mathbf{z}}_p; |\bar{\mathbf{z}}_r - \bar{\mathbf{z}}_p|; (\bar{\mathbf{z}}_r \odot \bar{\mathbf{z}}_p); \mathbf{h}_r^{(0)}; \mathbf{h}_p^{(0)}; \mathbf{c}_{rp}], \tag{19}$$

$$\ell_{rp}^{\text{main}} = \text{MLP}_{\text{edge}}(\mathbf{f}_{rp}) + \bar{\mathbf{z}}_r^\top \mathbf{B} \bar{\mathbf{z}}_p, \tag{20}$$

where $\mathbf{B}$ is a learned bilinear map and $\text{MLP}_{\text{edge}}$ is a 3-layer MLP ($512{\to}256{\to}1$ with GELU and dropout).

**(8) Prototype-conditioned contrastive score (logit-level).** We add a contrastive-like score term that increases similarity for candidate interacting pairs while penalizing similarity to prototype-based global "negatives". This term is injected directly into the logit (no auxiliary contrastive loss is optimized). Using $\ell_2$-normalized embeddings and temperature $\tau$, we define:

$$s_{\text{pos}} = \frac{\langle \bar{\mathbf{z}}_r, \bar{\mathbf{z}}_p \rangle}{\tau}, \tag{21}$$

$$s_{\text{negP}} = \log \sum_{k=1}^{K} \exp\left(\frac{\langle \bar{\mathbf{z}}_r, \mathbf{P}_{p,k} \rangle}{\tau}\right), \quad s_{\text{negR}} = \log \sum_{k=1}^{K} \exp\left(\frac{\langle \bar{\mathbf{z}}_p, \mathbf{P}_{r,k} \rangle}{\tau}\right), \tag{22}$$

$$s_{rp}^{\text{ctr}} = s_{\text{pos}} - \tfrac{1}{2}(s_{\text{negP}} + s_{\text{negR}}). \tag{23}$$

The final interaction logit is:

$$\ell_{rp} = \ell_{rp}^{\text{main}} + \lambda\, s_{rp}^{\text{ctr}}, \tag{24}$$

where $\lambda$ is a scalar weight. This component acts as a structured regularizer injected directly into the logit score.

**Ablation variants.** We evaluate three additional ProtoGNN variants (degree-gated propagation without prototypes, weighted bipartite propagation, and co-neighbor propagation). Full module definitions are provided in Appendix A.

## 3.4 Experimental Setup

**Evaluation protocol.** We use 5-fold stratified edge-level cross-validation on each dataset. In every fold, $20\%$ of labeled RNA–protein pairs form the *test* split. From the remaining $80\%$, we split off $10\%$ as an *inner validation* set (i.e., $8\%$ of the full data), leaving $72\%$ for training. All splits preserve class ratio via stratification.

**Leakage-free graph construction.** For graph-based variants (ProtoGNN), we construct the bipartite adjacency matrix $A \in \mathbb{R}^{|\mathcal{R}| \times |\mathcal{P}|}$ using training positives only in the current fold. No validation/test edges are used to build $A$. This ensures the message-passing structure does not leak label information.

**Training details.** All models are trained with AdamW (learning rate $10^{-3}$) for up to 50 epochs, with early stopping (patience $= 5$) using validation AUROC. We optimize BCEWithLogits with positive-class weight $w^+$:

$$\mathcal{L}_{\mathrm{BCE}} = -w^+ y \log \sigma(\ell) \ - \ (1 - y) \log\big(1 - \sigma(\ell)\big), \tag{25}$$

where $w^+ = \frac{N^-}{N^+}$ is computed from the training split of each fold. Unless stated otherwise, ProtoGNN uses latent dimension $d{=}256$, $L{=}2$ propagation layers, dropout $0.1$, and batch size $4096$. For the PairMLP baseline, we use the same projection dimension $d{=}256$, MLP hidden size $512$, and dropout $0.2$.

**Threshold tuning by MCC.** Although models output a probability $\hat{y} = \sigma(\ell)$, we do not fix the decision threshold at $0.5$. Instead, for each fold we select the threshold $\gamma^* \in [0, 1]$ that maximizes MCC on the validation split:

$$\gamma^* = \arg \max_{\gamma \in [0,1]} \mathrm{MCC}(\hat{y}(\gamma), y). \tag{26}$$

We then report test metrics using $\gamma^*$ for that fold, and finally summarize results as mean$\pm$std across folds.

**Metrics.** Let $TP, TN, FP, FN$ denote confusion-matrix counts. We report Accuracy (ACC), Sensitivity/Recall (SEN), Specificity (SPE), Precision (PRE), and MCC:

$$\mathrm{ACC} = \frac{TP + TN}{TP + TN + FP + FN}, \tag{27}$$

$$\mathrm{SEN} = \frac{TP}{TP + FN}, \qquad \mathrm{SPE} = \frac{TN}{TN + FP}, \tag{28}$$

$$\mathrm{PRE} = \frac{TP}{TP + FP}, \tag{29}$$

$$\mathrm{MCC} = \frac{TP \cdot TN - FP \cdot FN}{\sqrt{(TP+FP)(TP+FN)(TN+FP)(TN+FN)}}. \tag{30}$$

We additionally compute AUROC and AUPRC from logits as threshold-independent measures. We emphasize MCC because it captures performance on both classes using all four confusion-matrix terms, and is commonly used for RPI evaluation when balancing sensitivity and specificity is critical.

**Retrieval evaluation.** To assess RBP candidate prioritization, we additionally compute RNA-to-protein retrieval metrics (MRR, AP, Recall@K/Prec@K) by ranking *all* proteins for each query RNA using predicted interaction probability. We report macro-averaged retrieval scores over eligible RNAs; details are in Section 4.2 and Appendix D.

## 4 Results

### 4.1 Main comparison against published baselines

We evaluate on NPInter2 and RPI7317 under the 5-fold stratified edge-level protocol described in Section 3.4 and report mean$\pm$std across folds. Figures 1 and 2 summarize performance against published baselines reported on the same benchmarks. For our methods, PairMLP isolates the predictive signal from pretrained sequence embeddings alone, while ProtoGNN (contrastive) adds bipartite propagation, gated fusion, and prototype-conditioned scoring.

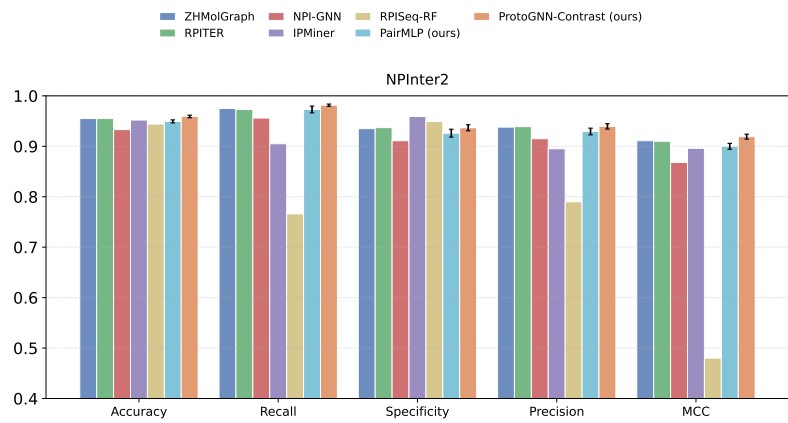

Figure 1: **Benchmark performance on NPInter2.** Comparison across ACC/Recall/Specificity/Precision/MCC.

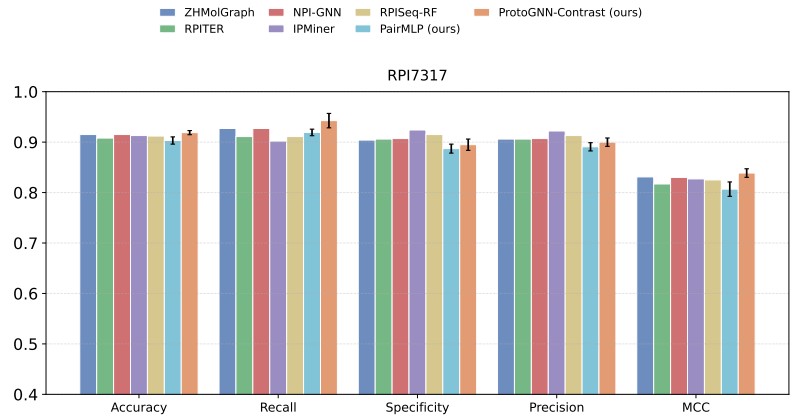

Figure 2: **Benchmark performance on RPI7317.** Comparison across ACC/Recall/Specificity/Precision/MCC.

**NPInter2.** PairMLP performs strongly using only pretrained embeddings (MCC $0.8999 \pm 0.0059$), indicating that fixed sequence representations already capture substantial interaction signal. ProtoGNN (contrastive) improves MCC to $0.9191 \pm 0.0049$, and the gains are reflected in a better overall balance across the reported confusion-matrix metrics (Figure 1).

**RPI7317.** On the harder multi-source dataset, ProtoGNN (contrastive) attains $0.8387 \pm 0.0085$ MCC and improves over PairMLP (MCC $0.8068 \pm 0.0143$). This suggests that graph refinement and prototype-conditioned scoring become more beneficial as interaction evidence becomes noisier and less homogeneous (Figure 2).

### 4.2 RNA-TO-PROTEIN RETRIEVAL (RBP IDENTIFICATION SETTING)

Beyond predicting a single RNA–protein pair, we evaluate an *RBP identification* setting: given an RNA query, ProtoGNN ranks *all* proteins by the predicted interaction score $s(r, p)$, and we measure how well known binders are prioritized in the top-$K$ list.

On NPInter2, ProtoGNN achieves macro-averaged retrieval performance of MRR $\approx 0.30$, Recall@10 $\approx 0.26$, Recall@20 $\approx 0.51$, and Recall@50 $\approx 0.73$ (mean over RNAs; consistent across folds). These results indicate that ProtoGNN can recover a meaningful fraction of known binders within small candidate lists and retrieve most known binders within the top 50–100. Precision at small $K$ remains modest (e.g., Prec@10 $\approx 0.17$), which is expected because many top-ranked can-

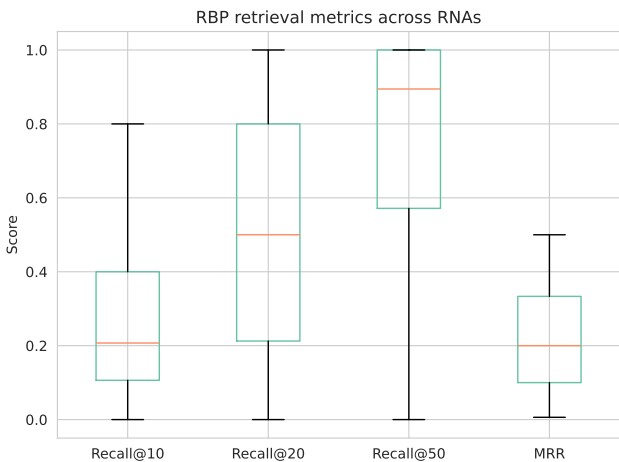

Figure 3: **Retrieval performance across query RNAs (NPInter2).** Boxplot summarizes per-RNA retrieval metrics, showing ProtoGNN's ability to prioritize candidate RBPs in a ranked list.

didates are *unlabeled* rather than verified negatives. Full retrieval distributions and case studies are reported in Appendix D.

### 4.3  ABLATION SUMMARY

We conducted ablations over ProtoGNN design choices (degree-gated propagation without proto-types, weighted bipartite propagation, co-neighbor propagation, and prototype-conditioned contrastive scoring). Overall, the prototype-conditioned contrastive variant was the most stable and achieved the strongest MCC across both datasets. Detailed module definitions and ablation results are provided in Appendix A and  B.

## 5  DISCUSSION

Our findings suggest two practical lessons for RPI prediction with pretrained sequence embeddings. First, embeddings alone already provide substantial discriminative signal: PairMLP achieves strong MCC, indicating that much of the performance comes from high-quality RNA/protein representations rather than complex feature engineering. Second, ProtoGNN yields consistent improvements over this embedding-only baseline, with larger gains on the noisier multi-source RPI7317 benchmark. Ablations indicate that streamlined bipartite propagation with adaptive raw→graph fusion is sufficient to refine node states, and that the prototype-conditioned, contrastive-inspired *logit augmentation* further improves separability beyond standard edge scoring. Accordingly, our RNA-to-protein retrieval evaluation shows ProtoGNN can prioritize candidate RBPs for a query RNA: it recovers a substantial fraction of known binders within the top 20–50 ranked proteins (Appendix D), providing a practical shortlist for follow-up validation. This connects the learned embedding space to a meaningful downstream use case rather than a pure score computation.

Our study has several limitations. Evaluation is restricted to two benchmarks with automatically generated negatives; while we follow the standard released protocol for comparability, random neg-ative sampling may include unknown true interactions and can affect absolute metrics. Moreover, edge-level CV does not fully reflect inductive discovery settings (unseen RNAs/proteins). Finally, retrieval evaluation is constrained by incomplete per-RNA labeling (many RNAs have positives but few/no labeled negatives), so we emphasize ranking-based metrics (MRR/Recall@K/AP) rather than per-RNA AUROC/AUPRC. Future work will (i) evaluate inductive splits and external datasets, (ii) explore biology-informed negatives and calibration, and (iii) analyze whether learned prototypes correspond to interpretable RNA/protein families to better support discovery.

## MEANINGFULNESS STATEMENT

Our work helps learn meaningful representations of life by turning pretrained sequence embeddings into interaction-aware representations that can be used not only for binary classification but also for biologically relevant retrieval. ProtoGNN injects relational structure from known RNA–protein interactions into fixed RNA and protein embeddings via bipartite refinement and prototype-conditioned global context. The resulting scores support RNA-centric ranking of candidate RNA-binding proteins (RBP identification), a common discovery workflow where models prioritize a shortlist for downstream validation. By emphasizing retrieval behavior and per-RNA heterogeneity, we evaluate whether representations capture functional interaction signals that generalize to practical candidate prioritization.

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

## A  PROTOGNN VARIANT DEFINITIONS

This appendix provides module-level definitions for the ProtoGNN ablation variants referenced in the Methods section. All variants share the same input embeddings (RNA-FM and ProtT5) and the same edge-level training protocol (AdamW, weighted BCE, 5-fold CV). Unless stated otherwise, the edge scoring head uses (i) interaction features $[\mathbf{u}; \mathbf{v}; |\mathbf{u} - \mathbf{v}|; \mathbf{u} \odot \mathbf{v}]$, (ii) a 3-layer MLP, and (iii) a bilinear term $\mathbf{u}^\top \mathbf{B} \mathbf{v}$.

### A.1  VARIANT (I): DEGREE-GATED PROPAGATION WITHOUT PROTOTYPES

**ProtoGNN-DegGate** removes the prototype construction and the prototype-conditioned contrastive augmentation. It retains: (a) bipartite propagation, (b) degree-gated raw→graph fusion, and (c) an edge-specific raw→graph gate.

**Propagation and degree-gated fusion.**  We compute $L$ layers of LightGCN-style bipartite updates using symmetric normalization as in Eq. (5–7) of the main text, producing propagated states $\mathbf{h}_r^{(L)}$ and $\mathbf{h}_p^{(L)}$. We then fuse raw and propagated representations by a learned degree gate:

$$g_r = \sigma(\mathrm{MLP}_r(\log(1 + \deg_r))), \qquad g_p = \sigma(\mathrm{MLP}_p(\log(1 + \deg_p))), \tag{31}$$

$$\mathbf{z}_r = \mathbf{h}_r^{(0)} + g_r \odot (\mathbf{h}_r^{(L)} - \mathbf{h}_r^{(0)}), \qquad \mathbf{z}_p = \mathbf{h}_p^{(0)} + g_p \odot (\mathbf{h}_p^{(L)} - \mathbf{h}_p^{(0)}). \tag{32}$$

**Type-aware global core (no prototypes).**  Instead of prototypes, we compute a lightweight global core vector by attention pooling over node embeddings:

$$w_r = \mathrm{softmax}(\mathbf{a}_r^\top \mathbf{z}_r), \qquad w_p = \mathrm{softmax}(\mathbf{a}_p^\top \mathbf{z}_p), \tag{33}$$

$$\mathbf{c} = \mathrm{Fuse}\left( \left[ \sum_r w_r(r)\mathbf{z}_r \; ; \; \sum_p w_p(p)\mathbf{z}_p \right] \right) \in \mathbb{R}^d, \tag{34}$$

where $\mathbf{a}_r, \mathbf{a}_p$ are learned vectors and $\mathrm{Fuse}(\cdot)$ is a small MLP.

**Pair-gated fusion and edge logit.**  For candidate edge $(r, p)$, we compute a scalar gate $a_{rp} \in (0, 1)$ from raw interaction features:

$$a_{rp} = \sigma(\mathrm{MLP}_{\mathrm{gate}}([\mathbf{h}_r^{(0)}; \mathbf{h}_p^{(0)}; |\Delta|; \odot])), \tag{35}$$

$$\bar{\mathbf{z}}_r = \mathbf{h}_r^{(0)} + a_{rp}(\mathbf{z}_r - \mathbf{h}_r^{(0)}), \quad \bar{\mathbf{z}}_p = \mathbf{h}_p^{(0)} + a_{rp}(\mathbf{z}_p - \mathbf{h}_p^{(0)}). \tag{36}$$

The final logit is:

$$\ell_{rp} = \mathrm{MLP}_{\mathrm{edge}}([\bar{\mathbf{z}}_r; \bar{\mathbf{z}}_p; |\bar{\mathbf{z}}_r - \bar{\mathbf{z}}_p|; \bar{\mathbf{z}}_r \odot \bar{\mathbf{z}}_p; \mathbf{h}_r^{(0)}; \mathbf{h}_p^{(0)}; \mathbf{c}]) + \bar{\mathbf{z}}_r^\top \mathbf{B} \bar{\mathbf{z}}_p. \tag{37}$$

### A.2  VARIANT (II): WEIGHTED BIPARTITE MESSAGE PASSING

**ProtoGNN-WeightedMP** replaces uniform (degree-normalized) propagation with an edge-weighted bipartite operator. For each observed training edge $(r, p)$ we compute a sigmoid attention weight:

$$w_{rp} = \sigma\left( \frac{\langle \mathbf{W}_q \mathbf{h}_r, \mathbf{W}_k \mathbf{h}_p \rangle}{\sqrt{d}} \right). \tag{38}$$

Messages are aggregated with per-node normalization:

$$\mathbf{m}_r = \frac{\sum_{p \in \mathcal{N}(r)} w_{rp} \mathbf{W}_v \mathbf{h}_p}{\sum_{p \in \mathcal{N}(r)} w_{rp} + \epsilon}, \qquad \mathbf{m}_p = \frac{\sum_{r \in \mathcal{N}(p)} w_{rp} \mathbf{W}_v' \mathbf{h}_r}{\sum_{r \in \mathcal{N}(p)} w_{rp} + \epsilon}. \tag{39}$$

Each layer applies a residual update with LayerNorm:

$$\mathbf{h}_r \leftarrow \mathrm{LN}(\mathbf{h}_r + \phi(\mathbf{m}_r)), \qquad \mathbf{h}_p \leftarrow \mathrm{LN}(\mathbf{h}_p + \phi(\mathbf{m}_p)), \tag{40}$$

where $\phi$ is GELU with dropout. The edge scorer matches the main model (MLP + bilinear) and uses the same global core vector.

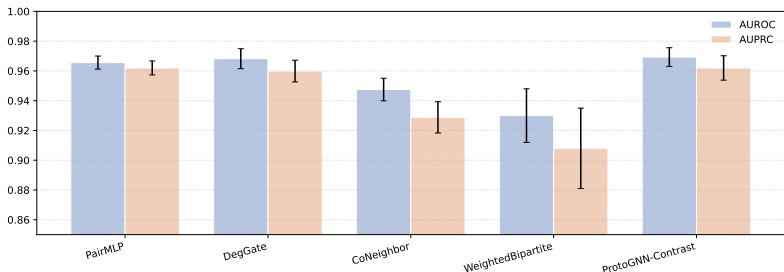

Figure 4: **AUROC/AUPRC across variants.** Threshold-free performance for PairMLP and ProtoGNN variants (mean±std over 5 folds).

### A.3 VARIANT (III): CO-NEIGHBOR (1-HOP + 2-HOP) PROPAGATION

**ProtoGNN-CoNeighbor** augments bipartite propagation with a 2-hop "co-neighbor" signal. Let $\tilde{\mathbf{A}}$ be the (optionally edge-dropped) training adjacency and $\mathbf{D}_r, \mathbf{D}_p$ the degree matrices. We compute:

$$\mathbf{m}_r^{(1)} = \mathbf{D}_r^{-1/2}\tilde{\mathbf{A}}\mathbf{D}_p^{-1/2}\mathbf{h}_p, \qquad \mathbf{m}_p^{(1)} = \mathbf{D}_p^{-1/2}\tilde{\mathbf{A}}^{\top}\mathbf{D}_r^{-1/2}\mathbf{h}_r, \tag{41}$$

$$\mathbf{m}_r^{(2)} = \mathbf{D}_r^{-1/2}\tilde{\mathbf{A}}\mathbf{D}_p^{-1/2}\mathbf{m}_p^{(1)}, \qquad \mathbf{m}_p^{(2)} = \mathbf{D}_p^{-1/2}\tilde{\mathbf{A}}^{\top}\mathbf{D}_r^{-1/2}\mathbf{m}_r^{(1)}. \tag{42}$$

We update node states using both 1-hop and 2-hop messages:

$$\mathbf{h}_r \leftarrow \text{LN}\left(\mathbf{h}_r + \text{MLP}_r([\mathbf{m}_r^{(1)}; \mathbf{m}_r^{(2)}])\right), \quad \mathbf{h}_p \leftarrow \text{LN}\left(\mathbf{h}_p + \text{MLP}_p([\mathbf{m}_p^{(1)}; \mathbf{m}_p^{(2)}])\right). \tag{43}$$

The remainder of the model (degree-gated fusion and edge scoring) matches the main architecture, but this block explicitly injects 2-hop co-neighborhood structure.

### A.4 THRESHOLD-FREE EVALUATION (AUROC/AUPRC)

MCC depends on a decision threshold (selected per fold by validation MCC). To provide a complementary threshold-independent view, Figure 4 reports AUROC and AUPRC for PairMLP and ProtoGNN variants. Across both datasets, the contrastive ProtoGNN variant achieves the most consistent AUROC/AUPRC among our tested designs, supporting that improvements are not solely an artifact of threshold selection.

## B ABLATION RESULTS

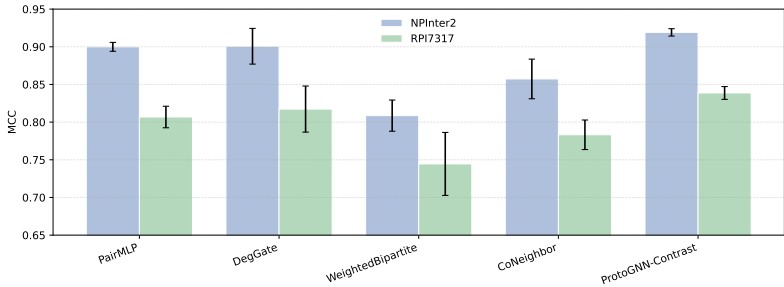

Figure 5: **Ablation (MCC).** Performance of ProtoGNN variants on NPInter2 and RPI7317 (mean±std over 5 folds).

## C ROBUSTNESS TO EDGE DROPOUT

We also examine sensitivity of ProtoGNN (contrastive) to sparse edge dropout on the training graph. Across the tested settings $p \in \{0.0, 0.1, 0.2\}$, performance remains stable, and a moderate dropout

rate is consistently competitive. In our sweep, NPInter2 achieves its best performance at $p = 0.2$ (MCC $0.9191 \pm 0.0049$), while RPI7317 achieves its best performance at $p = 0.1$ (MCC $0.8387 \pm 0.0085$). This suggests that mild edge-level regularization can help mitigate noise without degrading discriminative performance.

# D    RNA-TO-PROTEIN RETRIEVAL EVALUATION

## D.1    MOTIVATION

A brief summary of retrieval results is included in Section 4.2; this appendix provides full distributions, diagnostics, and qualitative case studies. Beyond edge-level classification (predicting whether a specific RNA–protein pair interacts), we evaluate *retrieval* performance in an RBP identification setting: given an RNA as a query, the model ranks candidate proteins by predicted interaction score. This mirrors practical usage in candidate prioritization, where a short list of likely RBPs is proposed for follow-up validation.

## D.2    PROTOCOL

For each 5-fold cross-validation split, we train ProtoGNN on the training edges (constructing the bipartite graph using training positives only, as in the main experiments). For evaluation, we consider each RNA that appears in the test split with at least one positive test interaction. For each such RNA $r$, we score candidate proteins $p$ using the trained model and rank proteins by predicted interaction probability $s(r, p)$.

## D.3    METRICS

We report standard retrieval metrics computed per RNA and macro-averaged across RNAs: *Recall@K* and *Precision@K* measure the fraction of true interacting proteins recovered within the top-$K$ ranked candidates and the purity of the top-$K$ list, respectively. *MRR* (mean reciprocal rank) captures how early the first true interactor appears in the ranked list, and *AP* (average precision) summarizes ranking quality across all positives for a query RNA.

## D.4    RESULTS AND INTERPRETATION

On NPInter2, ProtoGNN achieves macro-averaged retrieval performance of MRR $\approx 0.30$, Recall@10 $\approx 0.26$, Recall@20 $\approx 0.51$, and Recall@50 $\approx 0.73$ (averaged over RNAs; consistent across folds). These results indicate that, for a typical query RNA, ProtoGNN retrieves a meaningful fraction of its known binders within a small candidate list, and recovers most known binders within the top 50–100. While recall is consistently strong, precision at small $K$ remains modest (e.g., Prec@10 $\approx 0.17$), indicating that the top-ranked lists are noisy. We also observe substantial heterogeneity across RNAs (Figure 6), suggesting that some RNAs admit very accurate prioritization while others remain challenging.

## D.5    WHY PER-RNA AUROC/AUPRC IS OFTEN UNDEFINED

In principle, one may evaluate *per-RNA* score separability by treating, for a fixed query RNA $r$, its labeled test pairs $\{(r, p_i, y_i)\}$ as a small binary classification problem and computing AUROC/AUPRC from the predicted probabilities $s(r, p_i)$. However, under edge-level cross-validation on the RP interaction files, many RNAs appear in the test split with only positive labels (or only negatives), because negatives are not exhaustively annotated per RNA. For such single-class RNAs, AUROC/AUPRC is undefined. In our NPInter2 experiments, only a small subset of RNAs satisfied the criterion of having both labeled positives and negatives in the test fold; furthermore, the number of negatives per RNA was often very small (e.g., 1–3), making the resulting AUROC/AUPRC estimates unstable. Therefore, we report per-RNA AUROC/AUPRC only as an auxiliary diagnostic on the eligible subset and do not use it as a headline metric.

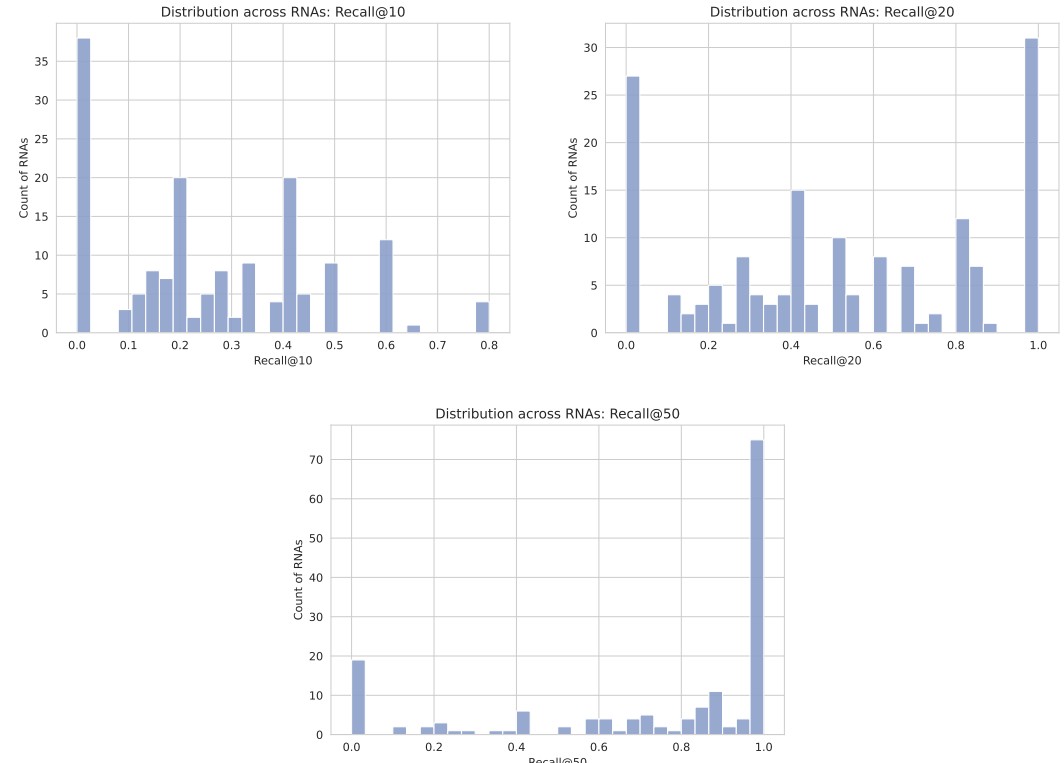

Figure 6: **Distribution of Recall@K across query RNAs.** Histograms show retrieval heterogeneity: some RNAs achieve high recall at small $K$, while others remain difficult.

D.6 CASE STUDY: CANDIDATE PRIORITIZATION FOR A SINGLE QUERY RNA

To mirror prior work that visualizes predictions for a single query (e.g., BioLLMNet case studies (Abir et al., 2025)), we perform an RNA-centric candidate prioritization analysis. Given a query RNA $r$, ProtoGNN scores *all* proteins $p$ in the dataset and ranks them by the predicted interaction probability $s(r, p)$. We then inspect the top-$K$ proteins and mark which of them correspond to labeled positives/negatives in the held-out test fold for that RNA.

Figure 8 shows an example for RNA ID 2669 on NPInter2 (one test fold). This RNA has 6 labeled interacting proteins in the test fold and no labeled negatives, hence per-RNA AUROC/AUPRC cannot be computed. Nevertheless, retrieval metrics remain meaningful: the best-ranked known binder appears at rank 1 (MRR= 1.0), and ProtoGNN retrieves 3/6 known binders in the top-10 (Recall@10= 0.5) and 5/6 in the top-50 (Recall@50≈ 0.83). Importantly, many of the remaining top-ranked proteins are *unlabeled* (i.e., absent from the test-fold edge list for this RNA) rather than confirmed negatives; thus, "no false positives" should be interpreted as "no top-ranked candidates were labeled as negative in the test split," not as verified non-interactions. Overall, this case study illustrates how ProtoGNN can be used for RBP candidate shortlist generation for follow-up validation.

While aggregate retrieval metrics summarize overall candidate-prioritization quality, they do not reveal *why* certain RNAs are easy or hard. Therefore, we include two additional qualitative examples: Figure 9 shows an RNA for which ProtoGNN produces a clean high-confidence shortlist (multiple labeled positives concentrated near the top ranks), whereas Figure 10 shows an RNA with poor early recall, where labeled positives appear later in the ranking and the top ranks are dominated by unlabeled candidates. These examples complement the distributional analyses and clarify typical success/failure modes.

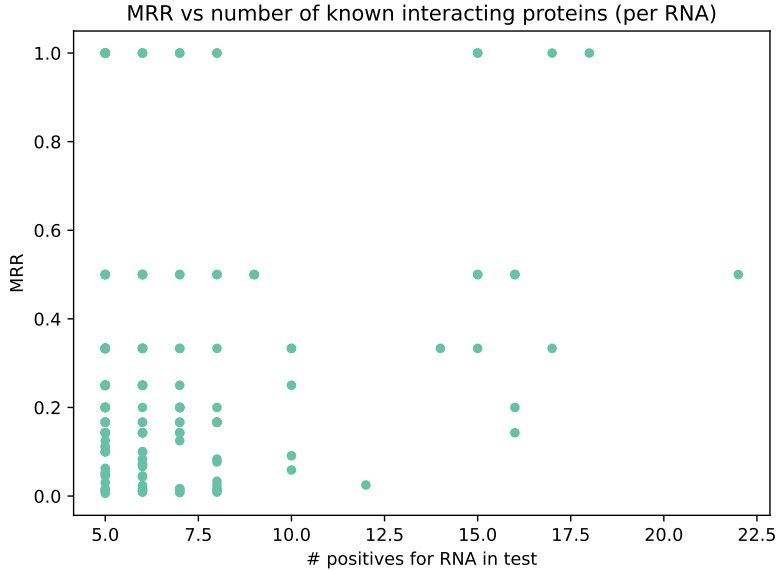

Figure 7: **Aggregate retrieval behavior.** MRR vs. number of known test positives per RNA, illustrating that ranking quality varies substantially and is only weakly explained by the number of known binders.

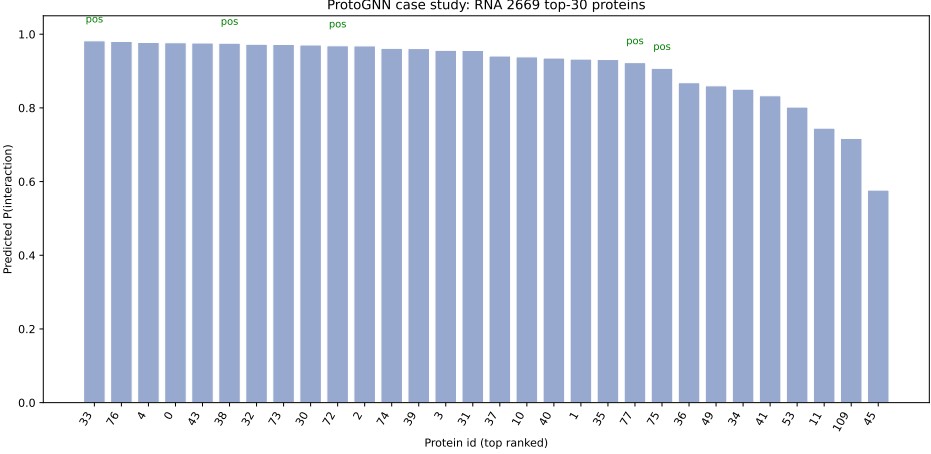

Figure 8: **Case study: RNA-centric protein ranking for RNA ID 2669 (NPInter2).** Bars show the predicted interaction probability for the top-30 ranked proteins. Proteins that are labeled as positives in the held-out test fold are annotated ("pos"). Unlabeled proteins are not evaluated in this per-fold case study but represent candidates for follow-up validation.

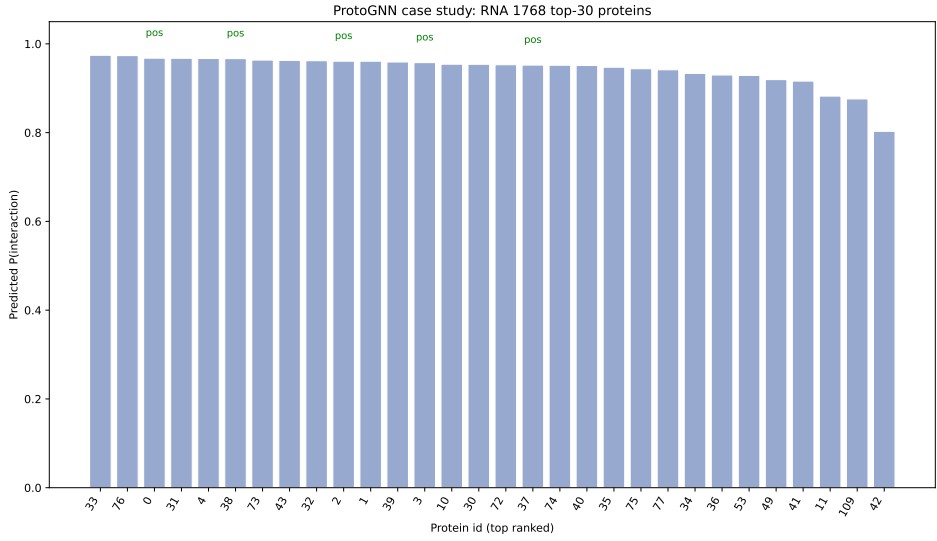

Figure 9: **Case study (easy RNA):** top-30 ranked proteins for an RNA with high retrieval performance (selected as the highest-MRR RNA among those with at least 5 positives in the test fold).

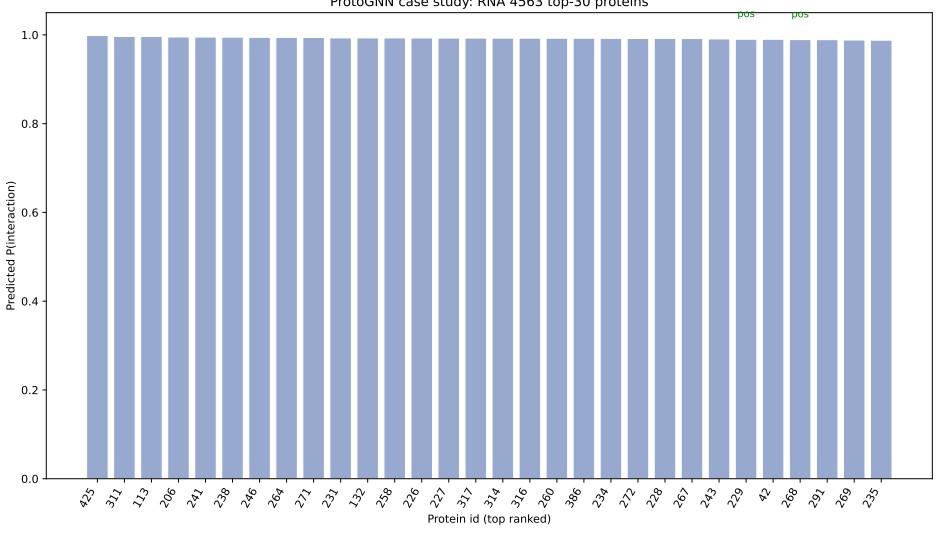

Figure 10: **Case study (hard RNA):** top-30 ranked proteins for an RNA with low Recall@10 (selected as the lowest-Recall@10 RNA among those with at least 5 positives in the test fold), illustrating failure modes and ranking noise.

