# OpenReview forum: "ProtoGNN: Prototype-Conditioned Graph Refinement for Meaningful RNA--Protein Interaction Representations"
_ICLR.cc/2026/Workshop/LMRL — Submitted to ICLR 2026 Workshop LMRL_

### Official Review · Reviewer_fYFj · 2026-02-20
**Incremental GNN refinement over pretrained embeddings for RPI with over-engineered components and insufficient evaluation rigor.**

**Rating:** 3
**Confidence:** 4

**Review:**

The paper proposes ProtoGNN, a bipartite GNN that refines fixed pretrained RNA (RNA-FM) and protein (ProtT5) embeddings for RNA–protein interaction prediction. The model adds degree-gated fusion, prototype-conditioned contrastive scoring, and pair-specific gates on top of standard bipartite message passing. A simple PairMLP baseline is introduced to quantify the contribution of graph structure. Results are reported on NPInter2 and RPI7317 under 5-fold edge-level CV, with retrieval metrics (MRR, Recall@K) for RBP identification.

Transparent baselining. The inclusion of PairMLP as an embedding-only baseline is commendable and reveals that pretrained embeddings alone achieve MCC 0.8999 on NPInter2 — making the actual contribution of ProtoGNN's graph machinery clearly quantifiable. This is the kind of honest experimental design more papers should adopt.

Retrieval evaluation. Moving beyond binary classification to RNA-to-protein retrieval is a useful framing that better reflects practical discovery workflows. The per-RNA heterogeneity analysis (Figures 6–10) and the honest discussion of why per-RNA AUROC is often undefined (Appendix D.5) show good experimental awareness.

Leakage-free graph construction. The authors correctly construct the bipartite adjacency using only training positives per fold, avoiding a common pitfall in GNN-based link prediction papers.

Ablation coverage. Four architectural variants are compared (DegGate, WeightedMP, CoNeighbor, Contrastive), with both threshold-dependent (MCC) and threshold-free (AUROC/AUPRC) metrics reported.

Clear mathematical presentation. The method section is well-organized with explicit equations for every component, making the architecture fully reproducible.

Marginal improvements over a trivial baseline. The core result is that ProtoGNN improves MCC from 0.8999 to 0.9191 on NPInter2 (+0.019) and from 0.8068 to 0.8387 on RPI7317 (+0.032). Given the standard deviations (±0.005 and ±0.009 respectively), the NPInter2 improvement is ~3σ but the absolute gain is tiny. The paper introduces degree gates, prototype attention, pair gates, bilinear terms, contrastive logit augmentation, edge dropout, and residual mixers — an enormous amount of architectural complexity for a 2-point MCC gain. No statistical significance tests (e.g., paired t-test across folds) are reported to confirm these differences are reliable.

Edge-level CV is a weak evaluation protocol, and the authors know it. The paper acknowledges in Section 5 that "edge-level CV does not fully reflect inductive discovery settings," yet proceeds to report only edge-level results. Under edge-level splits, both RNA and protein nodes appear in both train and test, meaning the GNN has seen all nodes during training. This is transductive leakage for any practical discovery scenario (predicting interactions for a new RNA). The fact that no inductive split is attempted — even as a supplementary experiment — is a significant omission for a paper claiming to support "biological discovery."

Comparisons to published baselines are not apples-to-apples. Figures 1–2 compare against "previously reported baselines" (ZHMolGraph, NPI-GNN, RPISeq-RF, etc.) from the literature, but the authors state these numbers are taken from prior publications. Differences in preprocessing, negative sampling, fold assignment, and threshold selection can easily account for 1–3 MCC points. The only fair comparisons are PairMLP vs. ProtoGNN variants, and those show marginal gains.

Prototype mechanism is not validated as meaningful. The paper claims prototypes provide "global context" and "improve separability," but never analyzes what the prototypes learn. Do RNA prototypes correspond to functional RNA classes (lncRNA, miRNA, etc.)? Do protein prototypes correspond to RBP families? The "meaningfulness statement" claims the representations are "useful beyond binary decisions," but the prototypes themselves are a black box. The retrieval evaluation tests the output scores, not the prototype representations.

Retrieval results are modest and hard to interpret. MRR ≈ 0.30 and Prec@10 ≈ 0.17 are low. The authors explain this by noting that top-ranked candidates are "unlabeled rather than verified negatives," which is fair — but this also means the retrieval evaluation is fundamentally uninterpretable. Without ground-truth negatives, Recall@K conflates model errors with annotation incompleteness. The case studies (Figures 8–10) are illustrative but anecdotal, and no comparison to PairMLP retrieval performance is provided, so we cannot assess whether graph refinement actually helps retrieval.

Random negative sampling is a known-fragile protocol. Both datasets use randomly paired RNA–protein pairs as negatives, matching 1:1 with positives. This inflates metrics for all methods (including baselines) because random pairs are trivially distinguishable from true interactions via embedding similarity alone — which explains why PairMLP already achieves MCC ~0.90. The paper does not test with biologically informed negatives (e.g., same-compartment non-interactors) that would more rigorously stress-test the model.

Overclaimed scope relative to LMRL workshop theme. The "Meaningfulness Statement" claims to "learn meaningful representations of life," but the paper's actual contribution is a modest architectural improvement on two standard benchmarks with known limitations. The representations are not shown to be meaningful in any biological sense — no GO enrichment, no functional clustering, no cross-dataset transfer.

---

### Official Review · Reviewer_SMsC · 2026-02-23
**Method slightly improving SOTA on protein-RNA interaction prediction but lacking justification of technical choices, context and clarity**

**Rating:** 5
**Confidence:** 3

**Review:**

## Overview
The paper addresses protein-RNA interaction prediction through graph neural networks. Several models represent RNAs and proteins as nodes in a graph and existing RNA-protein interactions as edges of the same graph. The current state-of-the-art model in the field, ZHMolGraph, proposes to leverage RNA and protein sequence foundation model embeddings as node features to improve prediction quality. Building on this success, ProtoGNN proposes to enhance graph neural networks such as the ones used in ZHMolGraph by introducing prototype learning, node degree-aware fusion, and pair-gated fusion for edge scoring

## General appreciation
This paper addresses an important, still under-addressed task: RNA-protein interaction prediction using deep learning and, more precisely, graph neural networks. By adding several components to the graph neural network architecture, it manages to outperform concurrent models, especially ZHMolGraph. However, several components are added to the architecture and not all are ablated in an isolated manner, which makes it difficult to know what is the crucial reason for the increase in performance. Besides, the writing lacks clarity, related work about GNN architectures, and motivation of the technical choices.

## Pros and cons
**Pros:**
* This paper addresses RNA-protein interaction prediction using deep learning, which is an important, underexplored problem
* The performance of ProtoGNN is improved over state-of-the-art methods such as ZHMolGraph
* The RNA-to-protein retrieval metric is relevant, and aligns with virtual screening-like metrics used in some binding affinity prediction models

**Cons:**
* The writing critically lacks clarity: the graph construction is not made explicit (we understand that nodes encode RNAs and proteins but this is not clearly stated, whereas this is not obvious since in the literature some papers use nodes for nucleotides/amino acids, some others for RNA/proteins, and some others do both), and the technical terms are not clearly explained (e.g., "type-specific prototypes" is not straightforward)
* No related work about graph neural networks, whereas the paper's innovation lies in GNN architecture rather than in biological task formulation (for instance, formulae 7 and 8 are reminiscent of the Graph convolution network (GCN) [1], which should be cited). Whether contributions are innovations in graph neural networks or applications to RNA-protein interaction of innovations proposed elsewhere in the graph neural network community (and in the latter case, where they were introduced) should be made explicit.
* The key technical choices (such as the use of prototypes) are poorly motivated
* The splitting is random, so the RNAs and proteins included in the train and test sets might be highly similar, potentially leading to inflated performance metrics
* Whereas the authors propose RNA-to-protein retrieval as a new metric and assess their model using this metric, they do not benchmark against concurrent models based on this metric
* No ablation assessing the impact of the degree-gated raw-graph fusion, all else being equal

[1] Kipf, Thomas N., and Max Welling. "Semi-supervised classification with graph convolutional networks." arXiv preprint arXiv:1609.02907 (2016).

---

### Meta-Review · Area_Chair_2wHi · 2026-02-28

**Recommendation:** Reject
**Confidence:** 3

**Metareview:**

The reviewers raised a number of flaws and missing references that need to be address before this could be accepted.

---

### Decision · Program_Chairs · 2026-03-02

**Decision:**

Reject

**Comment:**

Please see the meta-review.